# The linker histone H1–BRCA1 axis is a crucial mediator of replication fork stability

Meryem Ozgencil*, Arlinda Dullovi*, Romane Catherine Christiane Higos, Zuzana Hořejší, Roberto Bellelli

The maintenance of genome integrity relies on replication fork stabilization upon encountering endogenous and exogenous sources of DNA damage. How this process is coordinated with the local chromatin environment remains poorly defined. Here, we show that the replication-dependent histone H1 variants interact with the tumour suppressor BRCA1 in a replication stress–dependent manner. Transient loss of the replication-dependent histones H1 does not affect fork progression in unchallenged conditions but leads to the accumulation of stalled replication intermediates. Upon challenge with hydroxyurea, cells deficient for histone H1 variants fail to recruit BRCA1 to stalled replication forks and undergo MRE11-dependent fork resection and collapse, which ultimately leads to genomic instability and cell death. In summary, our work defines an essential role of the replication-dependent histone H1 variants in mediating BRCA1-dependent fork protection and genome stability.

## Introduction

The maintenance of genome stability depends on accurate and efficient DNA replication and is crucial for life in eukaryotes. Importantly, DNA damage caused by endogenous and external sources threatens the activity and processivity of the replication machinery and requires dedicated pathways for DNA repair (Cortez, 2019). For instance, persistent replication fork stalling, or replication stress, can be induced by non-canonical DNA structures (e.g., G-quadruplexes, G4s), reactive oxygen species, and other by-products of metabolic processes, as well as replication–transcription conflicts and oncogene activation in the early stages of tumorigenesis (Zeman & Cimprich, 2014). Consistently, replication stress is considered a major hallmark of cancer and is currently being exploited for therapeutic purposes (Macheret & Halazonetis, 2015).

An important mechanism required for genome stability is the protection of stressed replication forks by fork reversal and nascent strand protection, a process mediated by the concerted action of the RAD51 recombinase and the BRCA1-BRCA2 tumour suppressor pathway, in addition to a series of fork remodellers and processing enzymes (Quinet et al, 2017). How this process takes place in specific genomic regions and chromatin contexts remains poorly defined.

Gene expression programmes and cell identity require the regulated control of chromatin structure and epigenetic modifications. Thus, the histone core proteins H2A, H2B, H3, and H4 form an octameric complex that wraps ~146 bp of DNA into the nucleosome, the structural and functional unit of chromatin (Zhou et al, 2019). In addition, a chromatin architectural protein, the linker histone H1, binds to extra-nucleosomal DNA and the nucleosome particle to stabilize its folding into higher order chromatin structures, which are essential for regulation of genome function (Kowalski & Pałyga, 2016). Linker histones H1 are composed of a conserved central globular domain and two flexible, less conserved, N- and C-terminal tails. Seven histone H1 subtypes have been identified in somatic cells; of these variants, histones H1.1, H1.2, H1.3, H1.4, and H1.5 are synthesized during the S phase of the cell cycle and are known as replication-dependent variants (Kowalski & Pałyga, 2016). Although individual histone H1 subtypes present with different DNA affinity and chromatin distribution, their non-redundant roles in regulating epigenome function remain still debated (Prendergast & Reinberg, 2021). In mice, single or double H1 variant knock-out has no apparent phenotype because of the compensatory up-regulation of other subtypes (Fan et al, 2001). However, a combined lack of H1.2, H1.3, and H1.4 leads to overall reduction of histone H1 and aberrant differentiation, and results in embryonic lethality, pointing to a fundamental role of the replication-dependent histones H1 in modulating genome structure and function (Fan et al, 2003). Despite this, more recent work has challenged this knowledge and suggests different roles of histone H1 variants in regulating gene expression in specific chromatin and cellular contexts (Prendergast & Reinberg, 2021).

Although nucleosome dynamics and post-translational modifications are known to have a fundamental role in the DNA damage response (DDR) and DNA repair, the role of the histone H1 variants in the maintenance of genome stability remains poorly defined.

Centre for Cancer Cell and Molecular Biology, Barts Cancer Institute, Queen Mary University of London, London, UK

Correspondence: r.bellelli@qmul.ac.uk; zuzana.horejsi@uochb.cas.cz
Zuzana Hořejší's present address is IOCB Prague, Prague, Czech Republic
*Meryem Ozgencil and Arlinda Dullovi contributed equally to this work

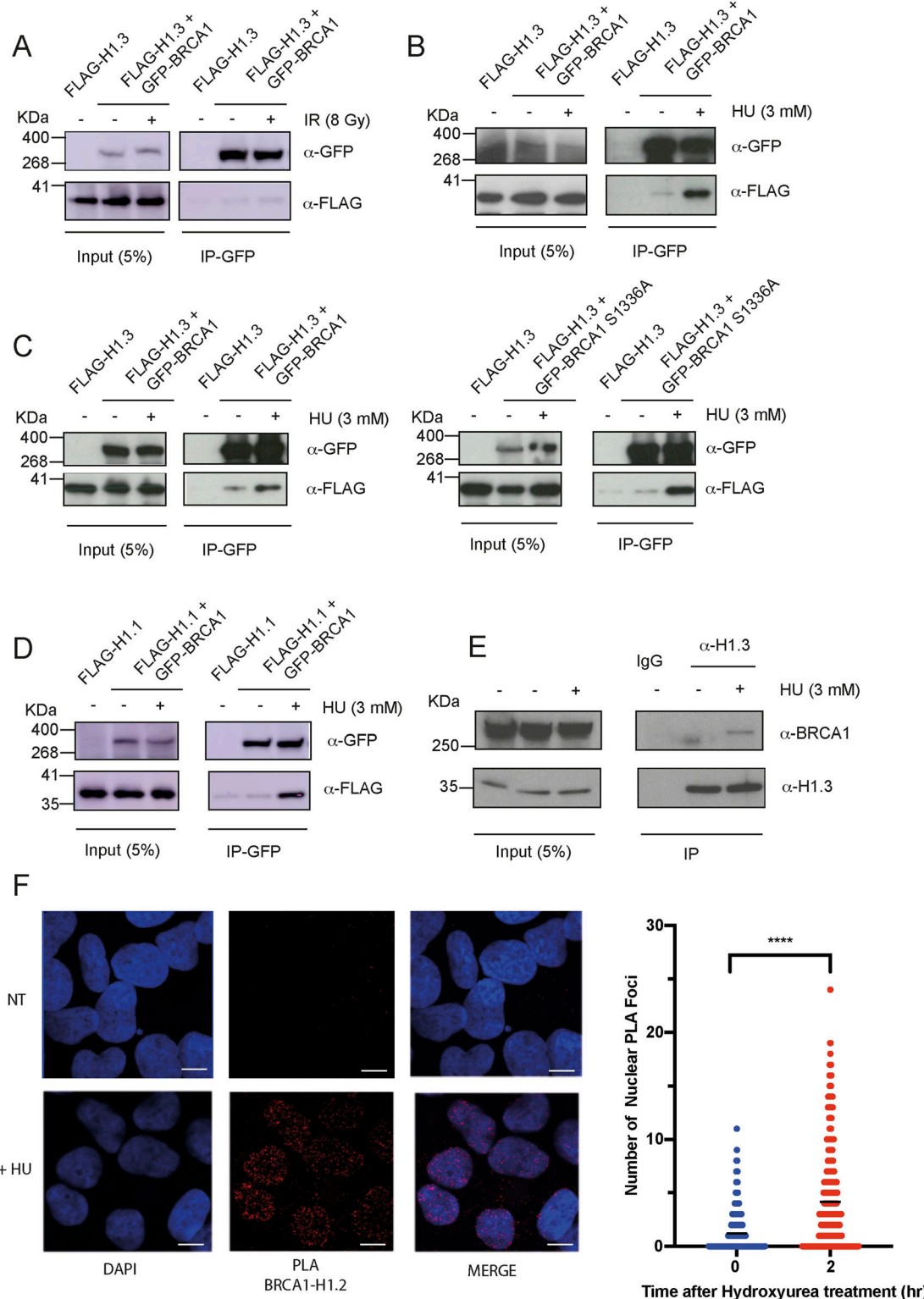

**Figure 1. BRCA1 interacts with the linker histone H1 upon treatment with hydroxyurea (HU).**
**(A)** Western blot analysis of GFP immunoprecipitations (IPs) from HEK293 cells transfected with GFP-BRCA1 and FLAG-H1.3 and treated or not with 10 Gy ionizing radiation. After SDS–PAGE and PVDF transfer, membranes were incubated with antibodies against the indicated proteins. **(B)** Western blot analysis of GFP IPs from HEK293 cells transiently transfected with GFP-BRCA1 and FLAG-H1.3 and treated or not with 3 mM HU for 2 h. After SDS–PAGE and PVDF transfer, membranes were incubated with antibodies against the indicated proteins. **(C)** Western blot analysis of GFP IPs from HEK293 cells transiently transfected with GFP-BRCA1 WT (left panel) or S1336A mutant (right panel), and FLAG-H1.3 and treated or not with 3 mM HU for 2 h. After SDS–PAGE, Western blotting was performed using antibodies against the indicated

Thorslund et al previously identified a role of histone H1 in mediating a ubiquitylation cascade at sites of DNA double-strand breaks (DSBs) required for their repair (Thorslund et al, 2015). The role of specific histone H1 variants in this process remains unknown. More recently, a specific role of the histone H1.2 variant has been described in modulating ATM signalling and DNA repair (Li et al, 2018).

Here, we show that the replication-dependent histone H1 variants are not required for normal replication fork progression under unchallenged conditions but are necessary for replication fork stability upon fork stalling and replication stress in a BRCA1-dependent manner. Thus, although loss of histone H1 does not affect activation of the DDR, ablation of replication-dependent histone H1 prevents BRCA1 recruitment to stalled replication forks and leads to MRE11-dependent resection of newly synthesized DNA. Loss of this protective mechanism leads to fork collapse, genetic instability, and finally cell death.

## Results

### BRCA1 directly interacts with the linker histone H1

We previously coupled peptide pull-downs and quantitative mass spectrometry to identify novel binding activities, within the DDR, dependent on CK2 phosphorylation (Horejsi et al, 2010, 2014; Dullovi et al, 2023). Interestingly, the linker histone variant H1.3 resulted as a top hit in our phospho-Ser1336 BRCA1 proteomic experiments (Dullovi et al, 2023). Furthermore, two other histone H1 variants, H1.5 and H1.2, were among the top 20 significantly enriched proteins (Dullovi et al, 2023). These data pointed to a physical interaction between BRCA1 and the DNA replication–dependent histone H1 variants. To test this hypothesis, we initially transfected HEK293 cells with plasmid vectors expressing GFP-BRCA1 and FLAG-H1.3 and performed GFP immunoprecipitations (IPs). Previous data have suggested a critical role of the linker histone H1 in promoting DNA repair via ubiquitin-dependent recruitment of the RNF168 ubiquitin ligase upon IR (ionizing radiation)-induced DSB formation (Thorslund et al, 2015). Thus, we initially performed co-IP experiments in cells subjected or not to 10 Gy IR. In these conditions, we observed a mild interaction between GFP-BRCA1 and FLAG-H1.3 that was only minimally increased upon IR (Fig 1A). Because the replicative linker histone variants are synthesized and incorporated into nascent chromatin during DNA replication, we reasoned that the interaction with BRCA1 might be more relevant in conditions of disrupted replication fork progression. Thus, we repeated our experiments in cells treated or not for 2 h with 3 mM hydroxyurea (HU). Strikingly, in these settings, we observed a strong increase in GFP-BRCA1 binding to FLAG-H1.3, suggestive of a replication stress–mediated interaction (Fig 1B).

Our proteomic experiment was performed using a peptide encompassing the CK2 substrate of BRCA1, serine 1336 (Dullovi et al, 2023); we therefore mutagenized S1336 into alanine (S1336A) and repeated co-transfection experiments. Strikingly, a S1336A mutant of full-length BRCA1 still interacted with FLAG-H1.3 in a replication stress–dependent manner (Fig 1C). Thus, although our data suggest that CK2-dependent phosphorylation of BRCA1 on S1336 might promote direct binding to histone H1 variants, other binding sites and/or indirect interactions also promote the replication stress–dependent interaction between BRCA1 and histone H1. In agreement with this, both the C-terminal portion, containing S1336, and the N-terminal portion, containing the ring domain, of BRCA1 interacted in co-transfection experiments with FLAG-H1.3 (Fig S1A).

In addition to H1.3, our proteomic experiments, performed in HeLa cell nuclear extracts, also identified the histone H1 variants H1.5 and H1.2 as potential interactors of BRCA1. Importantly, these specific histone H1 variants are highly expressed in cancer cells, suggesting that the interaction between BRCA1 and H1.3, H1.5, and H1.2 might depend on expression levels of these specific replicative histone H1 variants (Prendergast & Reinberg, 2021). With this in mind, we repeated co-transfection experiments with GFP-BRCA1 using FLAG-H1.1 and FLAG-H1.4, the two replication-dependent H1 variants that we did not identify in our mass spectrometry. Consistently with our hypothesis, we observed an interaction between GFP-BRCA1 and both FLAG-H1.1 and FLAG-H1.4 upon a HU challenge (Figs 1D and S1B). Moreover, using H1.2- and H1.3-specific antibodies we were able to observe that endogenous H1.2 and H1.3 interact with endogenous BRCA1 in a HU-dependent manner (Figs 1E and S1C). Importantly, using antibodies against endogenous BRCA1, H1.2, and H1.3 we also observed that the interaction between H1 and BRCA1 takes place specifically in the chromatin environment, in a HU-dependent manner, as visualized by the proximity ligation assay (PLA) (Figs 1F and S1D).

Finally, previous data have suggested a critical role of histone H1 ubiquitination in activating the DDR at sites of DSBs (Thorslund et al, 2015). Whether and how histone H1 ubiquitination promotes, directly or indirectly, the binding of BRCA1 to histone H1 in conditions of replication stress remained unknown. With this in mind, we transfected cells with siRNAs against RNF8 and analysed an interaction between endogenous BRCA1 and histone H1.3. Strikingly, knock-down of RNF8 strongly reduced an interaction between endogenous H1.3 and BRCA1. Thus, RNF8-dependent ubiquitination of histone H1 is crucial for the stable recruitment of BRCA1 to stalled replication forks (Fig S1E).

### Knock-down of histone H1 impairs BRCA1 recruitment to HU-induced DNA damage foci

The role of the histone H1 variants in DNA replication and DNA repair remains poorly defined. One caveat of research into histone

proteins. **(D)** Western blot analysis of GFP IPs from HEK293 cells transiently transfected with GFP-BRCA1 and FLAG-H1.1 and treated or not with 3 mM HU for 2 h. After SDS–PAGE, Western blotting was performed using antibodies against the indicated proteins. **(E)** IP of endogenous H1.3 from human U2OS cells treated or not with 3 mM HU for 2 h. After SDS–PAGE, Western blotting was performed using antibodies against the indicated proteins. **(F)** Left: representative pictures of proximity ligation assays (PLA) performed with antibodies against endogenous BRCA1 and H1.2 in U2OS cells treated or not with HU; scale bar = 10 $\mu$M. Right: quantification of PLA staining showing a number of PLA foci per nucleus in U2OS cells treated or not with HU.

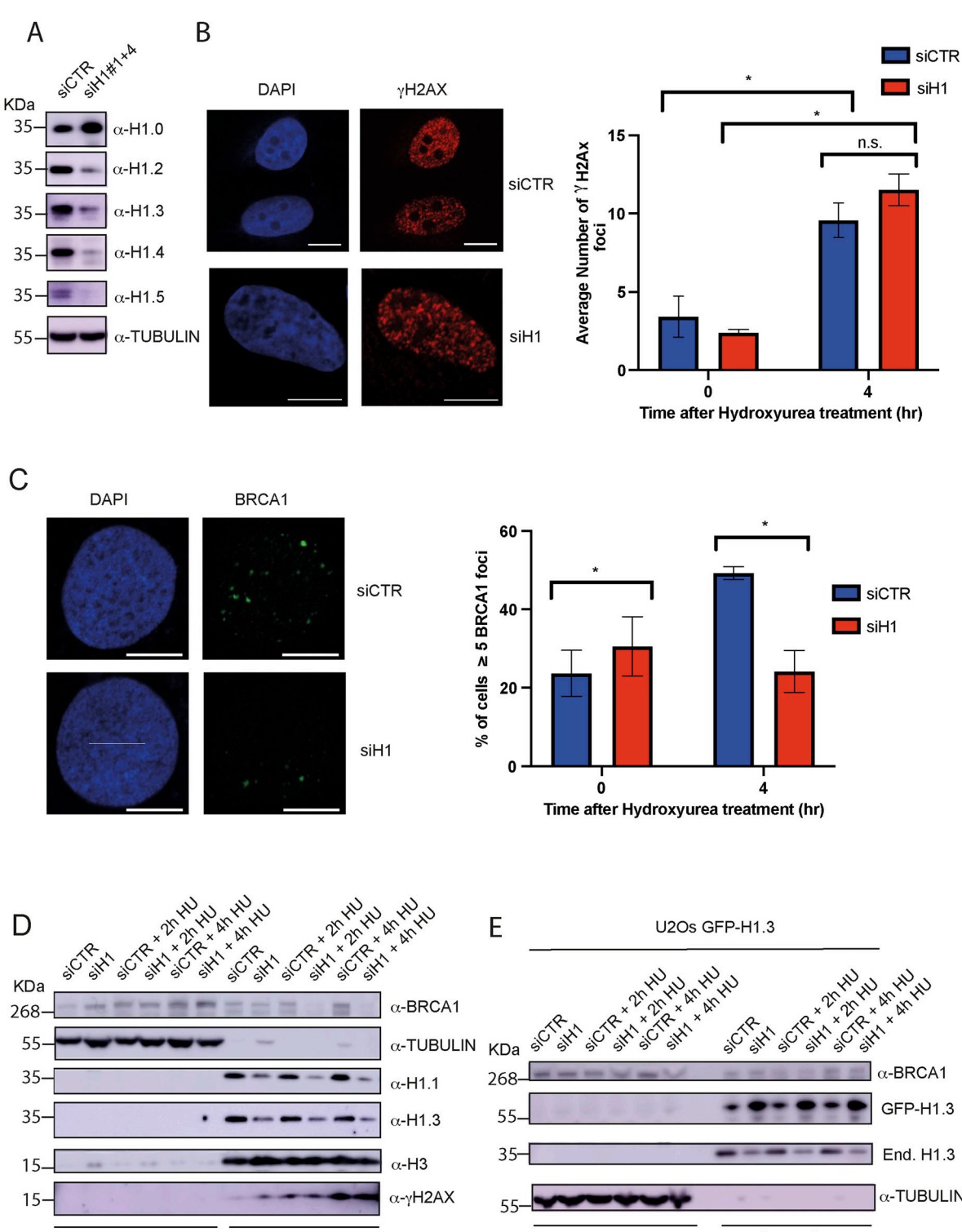

**Figure 2. Knock-down of histone H1 impairs chromatin recruitment of BRCA1 upon treatment with hydroxyurea (HU).**
**(A)** Western blot analysis of cell lysates from U2OS cells transfected with the indicated siRNAs. After SDS–PAGE and PVDF transfer, membranes were incubated with antibodies against the indicated proteins. Tubulin was used as a loading control. **(B)** Left: representative pictures from immunofluorescence staining for γH2AX in U2OS cells treated with the indicated siRNAs; scale bar = 10 μM. Right: bar graphs showing the average number of γH2AX foci in cells transfected with the indicated siRNAs; results are reported as the mean ± SD of three different experiments; unpaired *t* test analysis: *$P < 0.05$ and n.s., not significant. **(C)** Left: representative pictures from immunofluorescence staining for BRCA1 in U2OS cells treated with the indicated siRNAs; scale bar = 10 μM. Right: bar graphs showing the percentage of cells with more than five BRCA1 foci in the indicated conditions; results are reported as the mean ± SD of three different experiments; unpaired *t* test analysis: * $P < 0.05$ and n.s., not significant. **(D)** Western blot analysis of soluble and chromatin fractions from U2OS cells transfected with the indicated siRNAs and treated or not with HU for 2 and 4 h.

H1 variants is the compensatory increased expression of different H1 genes upon stable knock-out of one or two variants (Fan et al, 2001, 2003; Prendergast & Reinberg, 2021). With this in mind, we transiently silenced histone H1 variants using previously validated siRNAs (Thorslund et al, 2015 and Fig S2A). In our experimental settings, transfection with siH1#1 and siH1#4 resulted in the strongest preference for the replicative linker histone H1 variants H1.1–5 (Figs 2A and S2A). Consistently, combined transfection of cells with siRNAH1#1 and siRNAH1#4 caused the strongest silencing of H1.1–H1.5 without significantly affecting the replication-independent variant H1.0 (Fig 2A).

Given the increased interaction between BRCA1 and histone H1 upon replication stress, we reasoned that the replicative histones H1 might be required for activation of the DDR and/or the specific recruitment of BRCA1 to DNA damage foci upon HU-induced replicative damage. Similar to what previously reported during DSB repair, we observed that siRNA-mediated depletion of histone H1 did not hamper DDR activation upon HU treatment, as shown by phosphorylation of H2AX on Ser139 (Fig 2B). However, down-regulation of H1 strongly reduced BRCA1 focus formation 4 h after the addition of HU to the media (Fig 2C).

To alternatively visualize BRCA1 recruitment to sites of DNA damage, we then performed chromatin purification by CSK extraction in U2OS cells treated or not with siRNAs against the linker histones H1. Consistently with our immunofluorescence analysis, loss of H1 completely abrogated BRCA1 recruitment to chromatin (Fig 2D). Importantly, this was also associated with reduced chromatin levels of RAD51 (Fig S2B). Furthermore, although γH2AX levels were not affected, in agreement with our immunofluorescence analysis, we observed increased chromatin levels of replication protein A (RPA) (Fig S2C), pointing to the accumulation of ssDNA, likely from unscheduled fork resection and/or DNA strand breaks. Finally, in accordance with the specificity of our findings, the expression of a GFP-tagged and siRNA-resistant variant of H1.3 but not GFP-only was sufficient to rescue BRCA1 recruitment to chromatin upon HU treatment (Figs 2E and S2D). Of note, and in agreement with the compensatory effects played by the replication-dependent histone H1 variants, the over-expression of GFP-tagged and siRNA-resistant histones H1.1 and H1.4 also promoted chromatin recruitment of BRCA1 upon challenge with HU (Fig S2E). These data exclude a H1.3-specific mechanism for BRCA1 recruitment to stalled replication forks.

### Knock-down of histone H1 does not affect replication fork speed but leads to the accumulation of stalled replication forks

The role of the histone H1 variants in unchallenged DNA replication remains poorly defined. In preliminary experiments, we noted that transient siRNA-mediated knock-down of the replication-dependent H1 variants is associated only with a mild, not statistically significant, increase in the G2 phase of the cell cycle (Fig S3).

To monitor replication fork dynamics upon transient reduction of H1 variants, we initially analysed replication fork progression by the DNA fibre assay (Bellelli et al, 2018). Thus, we labelled cells transfected with siRNAs against histone H1 or CTR with the CldU and IdU halogenated nucleotides for two consecutive pulses of 20 min (Fig 3A). Using this approach, we found that transient reduction of histone H1 levels does not significantly affect replication fork progression (as shown by the fork rate) (Fig 3B). Importantly, transient knock-down of histone H1 was not associated with changes in interorigin distance, suggesting unaffected origin activation in these conditions (Fig 3C). We then analysed the symmetry of newly activated replication origins to look for signs of fork-stalling events (Fig 3D). Strikingly, transient loss of histone H1 was associated with an increased asymmetry of replication forks, thus suggesting increased fork stalling and/or defects in fork restart upon transient replication-stalling events (Fig 3E).

### Loss of histone H1 leads to increased resection of newly replicated DNA upon HU treatment

In addition to its well-established function in homologous recombination, the BRCA1 tumour suppressor has an important role in protecting newly replicated DNA from promiscuous nuclease activities (Schlacher et al, 2011, 2012; Quinet et al, 2017). Given the reduced recruitment of BRCA1 to chromatin upon HU treatment, we hypothesized that loss of H1, while not affecting overall fork progression, might lead to "un-protection" of transiently stalled replication forks, an event exacerbated upon HU treatment. To verify this hypothesis, we pulse-labelled cells with CldU and IdU (for 30′ and 30′) and then incubated them for 3 h in HU (Fig 4A). After DNA stretching and immunofluorescence staining, we then analysed the ratio between IdU and CldU as a measure of resection of newly replicated DNA upon HU-induced fork stalling. Strikingly, and in agreement with our hypothesis, knock-down of histone H1, in the presence of HU, led to resection of newly synthesized DNA, as visualized by the reduced IdU/CldU ratio (Fig 4A and B). These data suggest that loss of the histone H1 variants leads to fork "deprotection" via impaired recruitment of the BRCA1 tumour suppressor to transiently stalled replication forks.

To validate the BRCA1-specific function of histone H1 in fork protection, we then repeated DNA fibre experiments in BRCA1-mutant UWB1.289 ovarian cancer cells. In agreement with previous findings, HU treatment led to fork deprotection in UWB1.289 ovarian cancer cells as visualized by the reduced IdU/CldU ratio (Fig 4C). However, in this BRCA1-mutant cellular background, silencing of histone H1 did not further reduce IdU/CldU, pointing to an epistatic mechanism (Fig 4C). Importantly, siRNA-mediated knock-down of histone H1 led to fork deprotection in UWB1.289 cells complemented with wt BRCA1, pointing to a histone H1- and BRCA1-specific mechanism (Fig 4C). Finally, a similar epistatic mechanism was observed in U2OS cells transfected with siRNA against BRCA1 and histone H1 (Fig S4).

Tubulin and histone H3 were used as loading controls for soluble and chromatin fractions, respectively. **(E)** Western blot analysis of soluble and chromatin fractions from U2OS cells stably expressing siRNA-resistant GFP H1.3; cells were transfected with the indicated siRNAs and treated with HU for 2 and 4 h. Tubulin and histone H3 were used as loading controls for soluble and chromatin fractions, respectively.

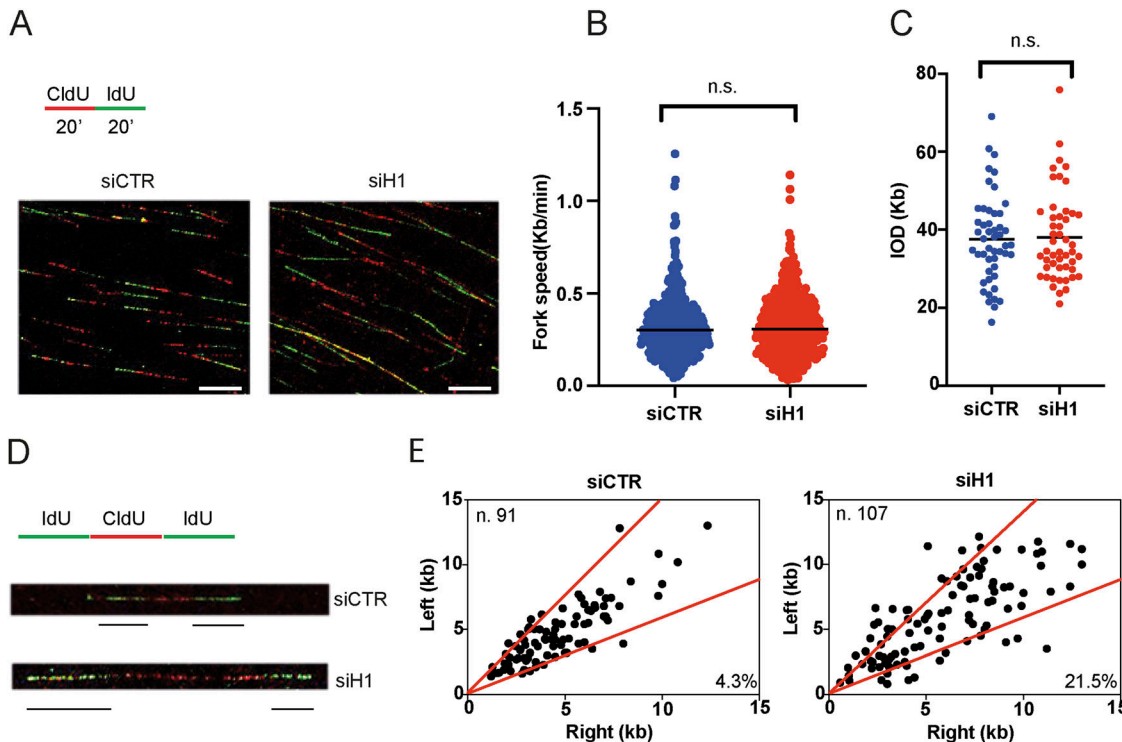

**Figure 3. Loss of the histone H1 leads to increased replication fork stalling.**
**(A)** Top: scheme of the nucleotide labelling strategy used in DNA fibre experiments. Bottom: representative pictures from DNA fibre immunofluorescence staining in U2OS cells treated with the indicated siRNAs; scale bar = 10 $\mu M$. **(B)** Bar graphs showing replication fork speed in U2OS cells transfected with siRNAs against replication-dependent histones H1 or CTR; data were obtained from technical duplicates from two different biological experiments; error bars ± SD are included; unpaired $t$ test analysis: n.s., not significant. **(C)** Bar graphs showing interorigin distance values in U2OS cells transfected with siRNAs against histone H1 or CTR; data were obtained from technical duplicates from two different biological experiments; error bars ± SD are included; unpaired $t$ test analysis: n.s., not significant. **(D)** Top: scheme of a newly activated replication origin, used for quantification of replication fork symmetry. Bottom: representative pictures of symmetric and asymmetric replication forks at newly activated origins from U2OS cells transfected with siRNAs against histone H1 or CTR. **(E)** Analysis of fork symmetry in U2OS cells transfected with the indicated siRNAs reported as left/right-moving fork ratio. The number of newly activated replication origins analysed is indicated together with the overall percentage of asymmetric forks.

Several research groups have previously shown that the major nuclease involved in resection of newly replicated DNA, upon loss of BRCA1-BRCA2, is MRE11 (Quinet et al, 2017). Consistently, treatment with the specific MRE11 inhibitor MIRIN has been previously shown to block resection of newly replicated DNA in BRCA-null cells (Schlacher et al, 2011, 2012). To validate the specificity of our findings and the role of MRE11 in this process, we then repeated our DNA fibre assay experiments with HU in the presence or absence of 50 $\mu M$ MIRIN (Fig 4D). Consistently with previous findings and our hypothesis, inhibition of MRE11 with MIRIN abrogated resection of stalled forks upon loss of histone H1. Thus, the linker histones H1 are required to prevent MRE11-dependent degradation of stalled replication forks.

**Loss of histone H1 leads to the accumulation of DNA breaks and reduced survival upon treatment with HU and PARP inhibitors**

DNA fibre experiments suggest that loss of histone H1 impairs BRCA1-mediated fork protection leading to uncontrolled MRE11-dependent fork resection. The increased chromatin levels and phosphorylation of RPA upon loss of histone H1 also suggested the premature collapse of replication forks. To validate this hypothesis, we treated cells with HU for 4 h, after histone H1 knock-down, and analysed the presence of DNA breaks by the alkaline comet assay. Strikingly, already 4 h after the addition of HU, we observed a strong increase in the length of the comet tails upon silencing of histone H1, thus pointing to premature fork breakage and collapse (Fig 5A).

We finally investigated the overall consequences of loss of histone H1 on genome stability and cell viability in conditions of disrupted replication fork progression. Our preliminary data suggest that loss of the replication-dependent histones H1 only minorly affects cell cycle progression (as shown by propidium iodide FACS analysis, Fig S3). Thus, to verify the effects of loss of histone H1 on genome stability we performed colony survival assays in cells transfected with siRNAs against histone H1, or control siRNAs, and treated or not with an increasing concentration of HU. Consistently with an important role in promoting genome stability and cell viability, siRNA-mediated knock-down of histone H1 led to a strong reduction in colony formation, when compared to untreated cells, at any HU concentration (Fig 5B). Importantly, and consistently with a BRCA1-specific mechanism, transient loss of histone H1 also led to increased sensitivity to PARP inhibitor olaparib (Fig 5C).

Hence, these data indicate that histone H1 variants are required to maintain replication fork stability upon transient or permanent

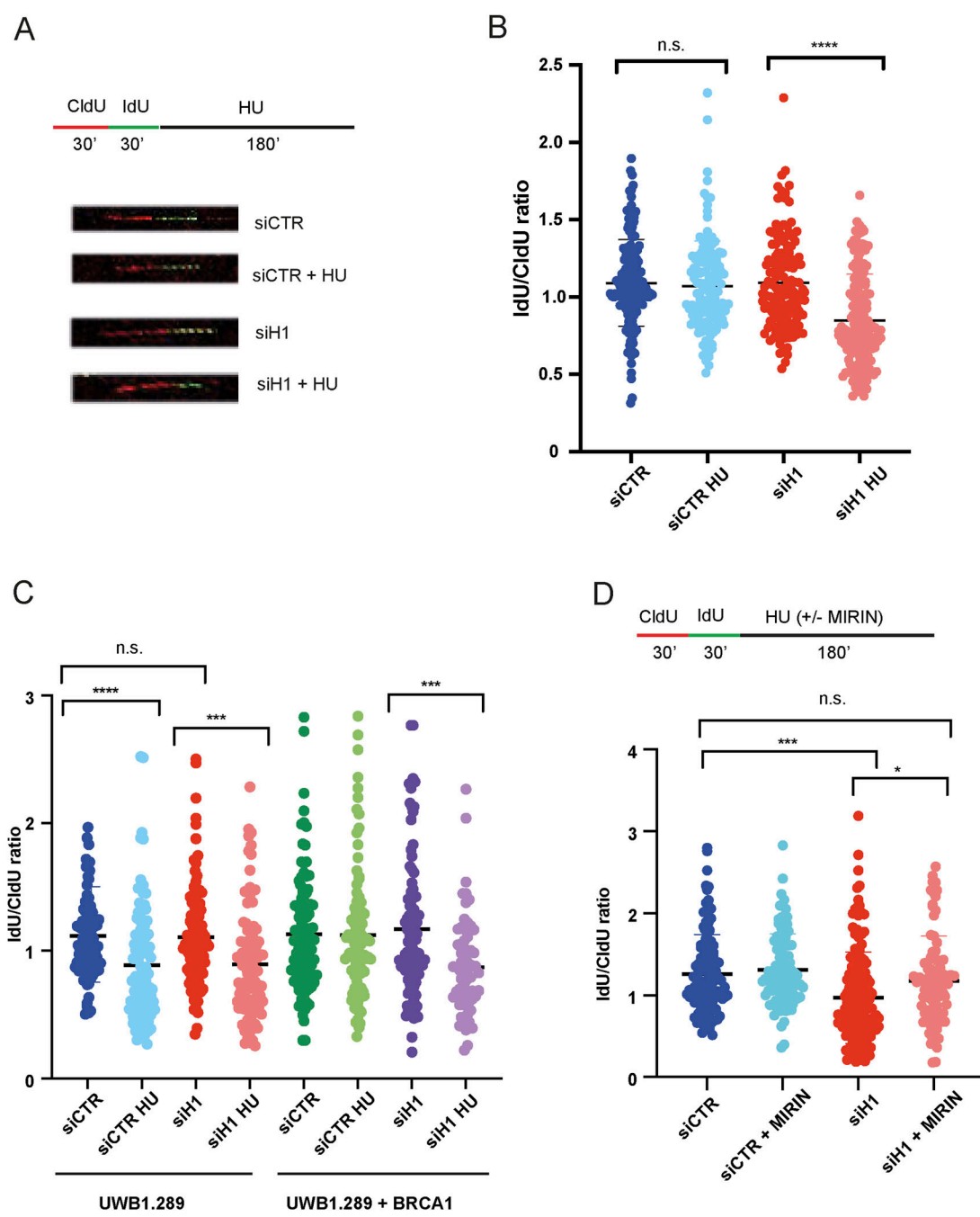

**Figure 4. Loss of histone H1 leads to MRE11-dependent fork resection upon HU treatment.**
**(A)** Top: scheme of the nucleotide labelling/HU treatment strategy used in DNA fibre resection assays. Bottom: representative pictures from DNA fibre immunofluorescence staining in U2OS cells treated with the indicated siRNAs and treated or not with HU (5 mM, 3 h). **(B)** Bar graphs showing the IdU/CldU ratio of individual replication forks in U2OS cells transfected with siRNAs against histone H1 or CTR and treated or not with HU (5 mM, 3 h); data were obtained from technical duplicates from two different biological experiments; error bars ± SD are included; unpaired *t* test analysis: ****$P < 0.0001$ and n.s., not significant. **(C)** Bar graphs showing the IdU/CldU ratio of individual replication forks in UWB1.289 BRCA1 mutant or UWB1.289 complemented with wt BRCA1, transfected with siRNAs against histone H1 or CTR, and treated or not with HU (5 mM, 3 h); error bars ± SD are included; unpaired *t* test analysis: ****$P < 0.0001$, ***$P < 0.001$, and n.s., not significant. **(D)** Bar graphs showing the IdU/CldU ratio of individual replication forks in U2OS cells transfected with siRNAs against histone H1 or CTR and treated with HU (5 mM 3, h) in the presence or absence of 50 µM MIRIN; data were obtained from technical duplicates from two different experiments; error bars ± SD are included; unpaired *t* test analysis: ***$P < 0.001$, *$P < 0.05$, and n.s., not significant.

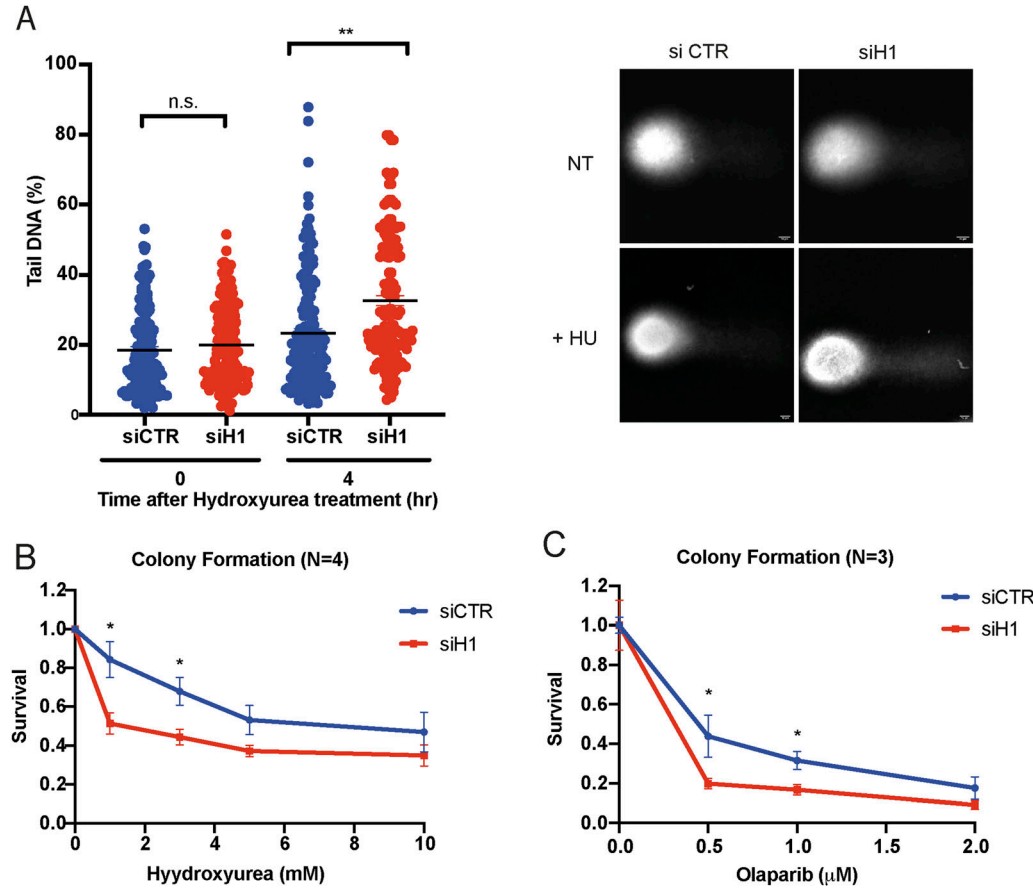

**Figure 5. Knock-down of histone H1 leads to the accumulation of DNA breaks and reduced viability upon HU treatment.**
**(A)** Left: quantification of alkaline comet assay experiments from U2OS cells transfected with the indicated siRNAs and treated or not with HU for 4 h. Data from three independent biological replicates are shown, and at least 150 cells were quantified for conditions. Error bars ± SD are included; unpaired *t* test analysis: \*\**P* < 0.01 and n.s., not significant. **(B)** Percentage survival of U2OS cells transiently transfected with the indicated siRNAs and treated with HU at the indicated doses. Data are the mean ± SD normalized to untreated cells (n = 4 biologically independent experiments); unpaired *t* test analysis: \**P* < 0.05. **(C)** Percentage survival of U2OS cells transiently transfected with the indicated siRNAs and treated with olaparib at the indicated doses. Data are the mean ± SD normalized to untreated cells (n = 3 biologically independent experiments); unpaired *t* test analysis: \**P* < 0.05.

stalling events and are thus required for genome integrity and cellular homeostasis.

## Discussion

By coupling peptide pull-downs and quantitative mass spectrometry, we previously identified and functionally validated novel interactions within the DDR (Horejsi et al, 2010, 2014; Dullovi et al, 2023). The linker histone H1 variants H1.3, H1.5, and H1.2 were identified as top hits in mass spectrometry experiments performed with a phosphopeptide encompassing the CK2 substrate S1336 of BRCA1 (Dullovi et al, 2023). However, the dynamics and functional relevance of this interaction remained unknown.

Here, we initially showed that the replication-dependent histone H1 interacts with BRCA1 in a replication stress–dependent manner. Importantly, treatment with IR only slightly increased the interaction between histone H1 and BRCA1, pointing to a specific replication stress–dependent mechanism. Furthermore, the fact that we observed an interaction between BRCA1 and histones H1.1 and H1.4 points to a lack of specificity towards specific histone H1 variants. We speculate that the expression levels of H1.3, H1.5, and H1.2, notoriously increased in cancer cells, might explain our initial mass spectrometry results, obtained using HeLa nuclear extracts (Prendergast & Reinberg, 2021).

Mutation of S1336 of full-length BRCA1 did not abrogate an interaction with FLAG-tagged H1.3; thus, other interaction sites are likely to be present within full-length proteins and further strengthen this interaction. Alternatively, other yet-to-be-identified indirect interactions might mediate and/or stabilize it. Consistently with this, silencing of the RNF8 ubiquitin ligase strongly reduced the binding of endogenous BRCA1 and histone H1.3 upon HU treatment. Structural reconstitution of BRCA1 in complex with a chromatin fibre containing both histone H1 and the core nucleosome might be required to solve this question. Recent work has indeed established a fundamental role of the obligate BRCA1 partner BARD1 in connecting H2A ubiquitination and new histone deposition to

homologous recombination in the S and G2 phases of the cell cycle (Becker et al, 2021). Consistently, BRCA1 recruitment to DSBs during the S phase requires recognition of H2A-Ub and H4 unmethylated at lysine 20 (H4K20me0), a classic marker of newly deposited histones and, as such, of post-replicative chromatin (Nakamura et al, 2019).

Independently from their role in canonical homologous recombination, BRCA1 and BRCA2 are required for the protection of newly replicated DNA from promiscuous nuclease activities, a function with important implications for both cancer biology and therapy (Schlacher et al, 2011; Ray Chaudhuri et al, 2016; Quinet et al, 2017). More specifically, the BRCA1-BRCA2 pathway promotes the stabilization of replication forks that underwent reversal upon encountering stalling events. Thus, upon fork stalling and reversal, the absence of BRCA1-BRCA2 leads to the unprotection of newly synthesized DNA- and MRE11-dependent fork resection (Kolinjivadi et al, 2017; Lemaçon et al, 2017; Mijic et al, 2017; Taglialatela et al, 2017); this finally results in MUS81-dependent cleavage of replication fork intermediates and the generation of double-strand breaks as a last resort rescue pathway (Lemaçon et al, 2017).

Transient knock-down of histone H1 abrogated HU-induced BRCA1 focus formation and chromatin recruitment. Consistently with fork deprotection, loss of histone H1 led to reduced IdU/CldU ratios in the presence of HU, a phenomenon suppressed by inhibition of MRE11. Thus, the replication-dependent histone H1 variants are required for BRCA1 recruitment at stalled replication forks and to prevent resection of reversed replication forks. The fact that we did not observe significant changes in fork speed upon knock-down of H1 suggests that the histone H1 variants are not required for fork reversal but for their protection (Quinet et al, 2017).

Transient knock-down of histone H1 was, however, associated with the accumulation of asymmetric forks in unchallenged conditions, suggesting the accumulation of stalled replication forks and/or defective restart of transiently stalled replication forks. Finally, and in agreement with an important role of histone H1 in replication fork stability, loss of histone H1 led to the accumulation of DNA breaks already 4 h after treatment with HU; this was associated with sensitivity to HU and the PARP inhibitor olaparib.

In summary, we established a novel link between the deposition and maintenance of chromatin structure during the S phase and the BRCA1 tumour suppressor pathway, which is required for replication fork stability and genome integrity. It is worth mentioning that a reduction of bulk H1 mRNA levels has been observed in up to 40% of ovarian adenocarcinomas (Medrzycki et al, 2012). As such, our work also suggests that loss of histone H1 might represent a therapeutic vulnerability and a source of genetic instability to be further investigated in human biology and disease.

# Materials and Methods

## Cell culture, siRNAs, and drug treatments

HEK293T and U2OS were grown in DMEM containing 10% FBS and 1% penicillin/streptomycin at 37°C in 5% $CO_2$. UWB1.289 and UWB1.289 reconstituted with WT BRCA1 (+BRCA1) were purchased from ATCC and grown in Mammary Epithelial Growth Medium, which is constituted of MEBM basal medium and SingleQuot additives (Mammary Epithelial Growth Medium Bullet Kit; CC-3150; Clonetics/Lonza) in 3% FBS and 1% penicillin/streptomycin at 37°C in 5% $CO_2$. For siRNA transfections, 6 µl of Lipofectamine RNAiMAX Reagent (Invitrogen) was diluted in 100 µl of DMEM, and separately, 400 nM of each siRNA was diluted in 100 µl of DMEM. After incubation for 5 min at room temperature, the two mixes were combined and incubated for a further 20 min at room temperature. The mixture was then added to the cells with 800 µl of DMEM + 10% FBS and incubated for 5 h before media were refreshed. The following siRNAs were used in this study:

siCTRL, 5′-GGGAUACCUAGACGUUCUATT-3′
siHistone H1(#1), 5′-GCUACGACGUGGAGAAGAATT-3′
siHistone H1(#4), 5′-CCUUUAAACUCAACAAGAATT-3′
siHistone H1(#5), 5′-CCUUCAAACUCAACAAGAATT-3′
siHistone H1(#9), 5′-CAGUGAAACCCAAAGCAAATT-3′
siRNF8, 5′-GGACAAUUAUGGACAACAA-3′
siBRCA1, 5′-CUAGAAAUCUGUUGCUAUG-3′

Cells were treated with 10 Gy IR or 3–5 mM HU (Sigma-Aldrich).

## Plasmids

cDNAs for histones H1.1, H1.3, and H1.4 were purchased from Dharmacon and initially cloned into pDONR221 (Invitrogen) by PCR. pDONR vectors were subsequently used for Gateway LR reactions with pDEST-FTF/FRT/TO and pDEST53 plasmids (Invitrogen) for the mammalian expression of FLAG- and GFP-tagged constructs, respectively. The GFP-BRCA1 expression plasmid was a gift from the Solomon laboratory (Morris & Solomon, 2004), whereas the S1336A mutation of BRCA1 was introduced using Q5 Site-Directed Mutagenesis Kit (NEB).

## Protein extracts, IP, and pull-down assays

Protein extracts were prepared in lysis buffer as previously described (von Morgen et al, 2017). For IPs and pull-downs, 2.5 mg of the whole-cell extract was incubated for 2 h at 4°C with 15 µl of M2 anti-FLAG agarose (Sigma-Aldrich), GFP-Trap agarose (ChromoTek), or protein G agarose (Cell Signaling) bound with primary antibodies. Beads were pelleted and washed three times in 20x bed volume of the lysis buffer. For immunoprecipitation, bound proteins were eluted by boiling in 2x LSB (Laemmli sample buffer) for 5 min. Inputs represent 5% of the extracts used for IP.

## Immunofluorescence

Cells transfected with the indicated siRNAs were grown on glass coverslips and treated with HU (3 mM) for 4 h before fixation in 4% paraformaldehyde (10 min at room temperature). After 3x washes in PBS, cells were permeabilized with 0.5% Triton–PBS for 10 min and incubated with primary antibodies for 1 h at room temperature. Coverslips were then washed three times with PBS and incubated with secondary antibodies for 1 h at room temperature in the dark. After three washes in PBS, coverslips were incubated in 0.2 µg/ml DAPI for 2 min at room temperature, washed, air-dried, and finally mounted onto glass slides using ProLong Gold Antifade Mountant (Invitrogen). Images were acquired with a Zeiss LSM 710 confocal

microscope using a 63× objective lens. BRCA1 and γH2AX focus quantification was performed using ImageJ. Cell nuclei were defined by thresholding the DAPI signal, and the regions of interest were saved to the ImageJ ROI manager. The BRCA1 and γH2AX signals were smoothed with the "Smooth" function, and the noise tolerance was adjusted using the "Find maxima" command to identify the BRCA1 and γH2AX foci. Foci for each preselected ROI were automatically counted, and results were presented as the sum of all pixels in the region on the ROI (RawIntDen). Because the pixel value is 255, the sum of all pixels was divided by 255 to quantify the number of BRCA1 and γH2AX foci in the defined nuclear region.

## PLA

U2OS cells grown on coverslips were transfected with siRNAs against histone H1 or control siRNAs and treated or not with HU for 2 h. Cells were then permeabilized with 0.2% Triton X-100 for 5 min at room temperature, and the PLA was performed using H1.2, H1.3, and BRCA1 antibodies and Duolink reagent (Sigma-Aldrich) according to the manufacturer's protocol.

## FACS analysis

siRNA-treated cells were harvested, washed with cold PBS, and fixed by adding ice-cold 70% ethanol dropwise while vortexing. After incubation on ice for 30 min, cells were washed with PBS and treated with 50 $\mu$g/ml RNase A for 30 min at 37°C. Finally, cells were stained with 20 $\mu$g/ml PI on ice for 30 min. Flow cytometry analysis was performed on a BD LSRFortessa cell analyser. Data were analysed with Flow Jo software using Cell Cycle Analysis Tool.

## Alkaline comet assay

The alkaline comet assay was performed using Reagent Kit for Single Cell Gel Electrophoresis Assay (Trevigen) with adaptations to the manufacturer's protocol. Cells were initially seeded and transfected with the indicated siRNAs and treated or not with 3 mM HU for 4 h. Cells were then suspended at $1 \times 10^5$ cells/ml, mixed with LMAgarose, and pipetted onto slides. Slides were incubated at 4°C in the dark for 1 h and subsequently incubated in lysis solution overnight at 4°C in the dark. After incubation in alkaline unwinding solution for 1 h at 4°C in the dark, the slides were placed in a horizontal electrophoresis tank containing alkaline electrophoresis solution and electrophoresis was carried out for 25 min at 30 V and 300 mA (at room temperature). The slides were washed twice for 5 min in water, then fixed in 70% ethanol for 5 min, dried at 37°C, and stored at 4°C. For DNA staining, slides were immersed in SYBR Gold in TE buffer (10 mM Tris–HCl, pH 7.5, and 1 mM EDTA) for 30 min at room temperature in the dark, washed with water, and finally dried at 37°C. Images were acquired using the Zeiss LSM 710 confocal microscope and 10× objective lens. A semi-automated image analysis was used to analyse 50 individual comets per gel to calculate the percentage of tail DNA (https://www.med.unc.edu/microscopy/resources/imagej-plugins-and-macros/comet-assay/).

## Chromatin fractionation

Cells in the mid-exponential phase of growth were washed once in PBS and lysed in ice-cold CSK (10 mM PIPES, pH 6.8, 100 mM NaCl, 300 mM sucrose, 1 mM MgCl$_2$, 1 mM EGTA, and 1 mM DTT) buffer containing 0.5% Triton X-100 (Pierce Biotechnology) and protease and phosphatase inhibitors (ROCHE) for 10 min on ice. Chromatin-bound and unbound proteins were separated by low-speed centrifugation (1,000$g$, 3 min at 4°C). The pellet (chromatin fraction) was washed in CSK 0.5% Triton and resuspended in 1X LSB. A total fraction was obtained by direct cell lysis in 1X LSB. For each fraction, protein amounts deriving from a comparable number of cells were analysed by SDS–PAGE and Western blotting.

## DNA fibre assay

DNA fibre assays were essentially performed as described in Bellelli et al (2018). Briefly, U2OS cells transfected with siRNAs against histones H1 or a control siRNA were pulse-labelled with 20 $\mu$M CldU for 20 min and subsequently with 200 $\mu$M IdU for 20 min. Cells were trypsinized, washed in PBS, and resuspended at a concentration of $5 \times 10^5$ in PBS. 2.5 $\mu$l of cell suspension was spotted on clean glass slides and lysed with 7.5 $\mu$l of 0.5% SDS in 200 mM Tris–HCl, pH 7.4, and 50 mM EDTA for 10 min at R.T. Slides were then tilted allowing a stream of DNA to run slowly down the slide, air-dried, and then fixed in methanol/acetic acid (3:1) for 15 min at R.T. After denaturation in HCl 2.5 M (30 min, R.T.), slides were blocked in 1% BSA/PBS and incubated with a rat anti-BrdU monoclonal antibody (1:1,000, overnight; Abcam) and a mouse anti-BrdU monoclonal antibody (1:500, 1 h, R.T.; Becton Dickinson). After washes in PBS, slides were incubated with Alexa Fluor 488 rabbit anti-mouse and Alexa Fluor 594 goat anti-rat antibodies (1:500, R.T.; Invitrogen) for 45 min and mounted in PBS/glycerol (1:1). Fibres were then examined using the Zeiss LSM 710 confocal microscope with 63x oil immersion objective and ImageJ software.

## Antibodies

Anti-FLAG (#2368) and γH2AX (#2577) antibodies were purchased from Cell Signaling Technology, whereas antibodies for BRCA1 (#sc-6954), H1.4 (#sc-393358), RNF8 (#sc-2714620), and GFP (#sc-9996) were from Santa Cruz Biotechnology. Antibodies against $\alpha$-tubulin (GTX628802) were obtained from GeneTex. Anti-RPA (ab2175), H1.0 (ab11079), H1.2 (ab17677), H1.3 (ab183736), and H1.4 (ab18208) were purchased from Abcam, whereas anti-H1.1 was from Insight Biotechnology (GTX117055). Finally, anti-RPA pS4+pS8 (A300-245A) antibodies were purchased from Bethyl, whereas anti-H3 antibodies were obtained from Sigma-Aldrich (05-928). Goat anti-mouse immunoglobulins/HRP (#P044701-2) and swine anti-rabbit immunoglobulins/HRP (P039901-2) were purchased from Dako. Goat anti-mouse IgG (H + L) secondary antibody, Alexa Fluor 488 (#A10680) and goat anti-rabbit IgG (H + L) cross-adsorbed secondary antibody, and Alexa Fluor 568 (#A11011) were purchased from Thermo Fisher Scientific.

### Clonogenic survival assays

For U2OS clonogenic survival assays, 1,000 cells were seeded per well in a six-well plate format in technical triplicate. HU and olaparib were added at different concentrations for 48 h. Cells were then grown for a further 10 d. Surviving colonies were stained using crystal violet, imaged, and counted using ImageJ. Treated cells were normalized to untreated samples.

### Statistical analysis

All statistical analyses were performed using GraphPad Prism 9 software. Statistical details for the experiments are provided in the figure legends. $P < 0.05$ was considered to be significant and classified by asterisks: $P < 0.05$ (*), $P < 0.01$ (**), $P < 0.001$ (***), and $P < 0.0001$ (****).

## Supplementary Information

## Acknowledgements

We would like to thank Ondrej Belan and Julian Stingele for critical reading of the article. This work was supported by Barts Charity Rising Star Program (to R Bellelli) and Wellcome Trust (200462/Z/16/Z) (to Z Hořejší).

### Author Contributions

M Ozgencil: data curation, formal analysis, investigation, and writing—review and editing.
A Dullovi: resources, data curation, formal analysis, and investigation.
RC Christiane Higos: investigation.
Z Hořejší: conceptualization, data curation, formal analysis, supervision, funding acquisition, investigation, methodology, project administration, and writing—original draft, review, and editing.
R Bellelli: conceptualization, data curation, formal analysis, supervision, funding acquisition, investigation, methodology, project administration, and writing—original draft, review, and editing.

### Conflict of Interest Statement

The authors declare that they have no conflict of interest.

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
