## [Reviewer comments · Life Science Alliance]

Life Science Alliance

The linker histone H1-BRCA1 axis is a crucial mediator of replication fork stability

Roberto Bellelli, Meryem Ozgencil, Arlinda Dullovi, Romane Higos, and Zuzana Hořejší

DOI: <https://doi.org/10.26508/lsa.202301933>

Corresponding author(s): Roberto Bellelli, Queen Mary University of London

Review Timeline:

Submission Date:	2023-01-19
Editorial Decision:	2023-02-09
Revision Received:	2023-06-05
Editorial Decision:	2023-06-07
Revision Received:	2023-06-13
Accepted:	2023-06-13

Scientific Editor: Novella Guidi

Transaction Report:

February 9, 2023

Re: Life Science Alliance manuscript #LSA-2023-01933

Dr. Roberto Bellelli
Queen Mary University of London
Barts Cancer Institute
Charterhouse Square
London EC1M 6BQ
United Kingdom

Dear Dr. Bellelli,

Thank you for submitting your manuscript entitled "The linker histone H1-BRCA1 axis is a crucial mediator of replication fork stability" to Life Science Alliance. The manuscript was assessed by expert reviewers, whose comments are appended to this letter. We invite you to submit a revised manuscript addressing the Reviewer comments.

Thank you for this interesting contribution to Life Science Alliance. We are looking forward to receiving your revised manuscript.

Sincerely,

B. MANUSCRIPT ORGANIZATION AND FORMATTING:

Reviewer #1 (Comments to the Authors (Required)):

In this manuscript, the authors show that histone H1 variants interact with BRCA1 in a replication-stress-dependent manner and Transient loss of the replication-dependent histones H1 leads to the accumulation of stalled replication intermediates. This is linked to BRCA1 recruitment failure leading to Mre11-dependent degradation of nascent DNA. This is an interesting work presenting original results. Additional experiments are required to consolidate the data presented.

Major critiques:

-It is unclear whether BRCA1-H1 interaction takes place on chromatin. This could be addressed by proximity ligation assay (PLA).

-The authors should check whether RAD51 chromatin binding is affected by the downregulation of histone H1

-The authors should verify whether the downregulation of H1 affects nascent DNA stability in one of the BRCA1-defective cancer cell models.

Minor critiques

-The colony survival assay in Fig 5B should be better explained. Is the downregulation of H1 shown here stable over several cell cycles?

Reviewer #2 (Comments to the Authors (Required)):

This is an interesting manuscript that identifies an interaction between histone H1 and BRCA1 and points to a role for H1 in fork protection. More in details, the authors found that BRCA1 interacts with the linker histone H1 variants upon treatment with HU. Upon knock-down of H1, BRCA1 fails to be recruited to chromatin despite normal activation of the DDR. The authors then elegantly use DNA fiber assay to study the dynamics of DNA replication forks upon siRNA of Histone H1 in the presence and absence of HU and demonstrate that loss of H1 leads to fork resection, dependent on MRE11. Overall, I think the experiments are well conducted and the findings will be of interest to the field. Thus, I am overall favorable for publication of this study. Having said that, some aspects could be improved.

Authors could investigate the pathways that support this interaction to strengthen their findings. Work from the Mailand lab showed that histone H1 ubiquitination is involved in DSB repair via RNF8 (Thorslund et al., Nature, 2015). How interaction between H1 and BRCA1 is modulated by this pathway ? and what about the intra-S-phase checkpoint ? (is interaction abrogated by treatment with ATRi ? or is it prevented by knock-down of RNF8 ?)

It's interesting that H1 knock-down does not change fork speed. I was wondering if reduction of histone H1 also affected the number of origins given the reduced chromatin compaction ?

Accordingly to authors model, siRNA of histone H1 should lead to a "BRCAness". This aspect is very exciting. Have authors tested sensitivity to PARP inhibitors ?

Reviewer #3 (Comments to the Authors (Required)):

In the present manuscript Ozgencil et al. report the identification of a novel and replication stress dependent interaction between BRCA1 and the linker histone H1 and propose a role for histone H1 in replication fork stability. The interaction was identified in a previous mass spec from the same authors and proposed to be dependent on CK2 phosphorylation. However, mutation of BRCA1 on this specific residue does not abrogate this interaction, suggesting the presence of multiple interaction sites.

Consistent with a functional role for this interaction silencing of histone H1 by siRNA prevented BRCA1 localization to sites of DNA damage while not preventing γ H2AX phosphorylation. Given the previously established link between BRCA1 and fork protection they show that loss of H1 induces fork degradation, all dependent on the MRE11 nuclease.

The work is interesting given the increased attention to histone H1 and its roles in regulating chromatin structure. The manuscript is well written and data are clearly presented. I have a few minor comments that authors should address to fully support publication.

It would be good to further strengthen the finding that this interaction and function does not depend on a specific histone H1 subtype. I understand that this issue is complicated by the different expression levels (in different cell lines) of the H1 variants but authors could strengthen this by:

- 1) analyse the interaction between BRCA1 and another histone H1 subtype, different from H1.3, at the endogenous levels (and/or at least in a different cell line ?).
- 2) The authors showed that expression of siRNA resistant GFP-H1.3 rescue BRCA1 recruitment. Could they see a similar effect with another histone H1 gene overexpression? this would strengthen their finding that this effect is not H1.3-specific.
- 3) IP with flag 1.4 seems weaker. Is this a wb exposure issue? a plasmid expression problem or might depend on different properties of H1.4 ?
- 4) are there other potential CK2 sites involved which might explain authors results? is this interaction reduced by CK2 inhibition for example? (this should be at least discussed)
- 5) Minor comment: Fig 2b. Contrast should be increased as γ H2AX staining is hardly visible.

Reviewer #1 (Comments to the Authors (Required)):

In this manuscript, the authors show that histone H1 variants interact with BRCA1 in a replication-stress-dependent manner and Transient loss of the replication-dependent histones H1 leads to the accumulation of stalled replication intermediates. This is linked to BRCA1 recruitment failure leading to Mre11-dependent degradation of nascent DNA. This is an interesting work presenting original results. Additional experiments are required to consolidate the data presented.

First of all, we would like to thank the reviewer for their positive comments on our work and the experimental suggestions. We believe we have addressed their comments in full:

Major

critiques:

-It is unclear whether BRCA1-H1 interaction takes place on chromatin. This could be addressed by proximity ligation assay (PLA).

To address the reviewer comment we have now performed PLA assays using antibodies against endogenous BRCA1 and H1.2 (see revised Fig. 1F) and H1.3 (see revised Fig. S1D). Our new data show that interaction between endogenous BRCA1 and H1 takes place on chromatin and is specifically induced by HU treatment.

-The authors should check whether RAD51 chromatin binding is affected by the downregulation of histone H1

To address the reviewer comment we have performed chromatin fractionation (upon silencing of histone H1 and treatment with HU) and analysed Rad51 levels (see revised Fig. S2B). Our new data show that loss of histone H1 impairs Rad51 chromatin loading upon treatment with HU. This is in agreement with the role of BRCA1 in promoting Rad51-dependent stabilization of stalled replication forks.

-The authors should verify whether the downregulation of H1 affects nascent DNA stability in one of the BRCA1-defective cancer cell models.

To address the reviewer comment we have now performed DNA fiber/fork protection assays in UWB1.289 ovarian cancer cells harbouring BRCA1 mutations. Importantly, we have also performed the same experiments in UWB1.289 complemented with WT BRCA1. As shown in our revised manuscript (see Fig. 4C) treatment with HU leads to a reduction of the IdU/CldU ratio in UWB1.289 cells, in agreement with the established role of BRCA1 in fork protection. However, this effect is not worsened by silencing of histone H1, pointing to an epistatic mechanism. In agreement with this, UWB1.289 complemented with wt BRCA1 showed restored fork protection, an effect that was lost upon silencing of histone H1. To further corroborate the genetic and mechanistic interaction between H1 and BRCA1 we have also performed co-silencing experiments in U2OS. As shown in revised Fig. S4, silencing of H1 and BRCA1 leads to a similar decrease in the IdU/CldU ratio (fork deprotection). Importantly this effect was not worsened by H1 and BRCA1 co-depletion, pointing to an epistatic mechanism.

Minor critiques

-The colony survival assay in Fig 5B should be better explained. Is the downregulation of H1 shown here stable over several cell cycles?

We apologize with the reviewer for the lack of clarity. Colony forming assays were done upon transient silencing of histone H1 (which is stable for at least 72 hours in our experiments). We clarified this in the revised text.

Reviewer #2 (Comments to the Authors (Required)):

We would like to sincerely thank the reviewer for their enthusiastic support. We have addressed their minor comments below.

This is an interesting manuscript that identifies an interaction between histone H1 and BRCA1 and points to a role for H1 in fork protection. More in details, the authors found that BRCA1 interacts with the linker histone H1 variants upon treatment with HU. Upon knock-down of H1, BRCA1 fails to be recruited to chromatin despite normal activation of the DDR. The authors then elegantly use DNA fiber assay to study the dynamics of DNA replication forks upon siRNA of Histone H1 in the presence and absence of HU and demonstrate that loss of H1 leads to fork resection, dependent on MRE11. Overall, I think the experiments are well conducted and the findings will be of interest to the field. Thus, I am overall favorable for publication of this study. Having said that, some aspects could be improved.

Authors could investigate the pathways that support this interaction to strengthen their findings. Work from the Mailand lab showed that histone H1 ubiquitination is involved in DSB repair via RNF8 (Thorslund et al., Nature, 2015). How interaction between H1 and BRCA1 is modulated by this pathway ? and what about the intra-S-phase checkpoint ? (is interaction abrogated by treatment with ATRi ? or is it prevented by knock-down of RNF8 ?)

To strengthen our manuscript and address the reviewer comments we have now further investigated the mechanistic basis of the H1-BRCA1 interaction. We now show that interaction between H1 and BRCA1 takes place specifically on chromatin (Revised Fig. 1F and S1D) upon HU treatment and requires both the C-terminal domain (containing the CK2 phosphorylation site S1336) and the ring domain of BRCA1 (Revised Fig. S1A). These data suggest that interaction between H1 and BRCA1 involves multiple sites. In agreement with this and to address the reviewer comment we now also show that transient silencing of RNF8 strongly reduces interaction between endogenous H1 and BRCA1 (revised Fig. S1E). This data points to a critical role for H1 ubiquitination in regulating binding to and recruitment of BRCA1 to stressed replication forks. Treatment with ATR inhibitors is known to reduce fork speed and induce fork stalling while affecting hundreds of substrates within the DDR. We believe it would be very difficult to raise any conclusions based on the use of such inhibitors. We hope the reviewer will agree with our conclusions.

It's interesting that H1 knock-down does not change fork speed. I was wondering if reduction of histone H1 also affected the number of origins given the reduced chromatin compaction ?

We thank the reviewer for the experimental suggestion. We have now analysed inter origin distance values upon silencing of H1 by DNA fiber assay and found that downregulation of the replication-dependent histone H1 variants does not affect origin activation (as shown by absence of IOD changes, see revised Fig. 3C)

Accordingly to authors model, siRNA of histone H1 should lead to a "BRCAness". This aspect is very exciting. Have authors tested sensitivity to PARP inhibitors ?

We thank the reviewer for the comment and experimental suggestion. Indeed, we now show that transient silencing of histone H1 leads to sensitization to PARP inhibitors, in agreement with our "H1-BRCA1 axis" model (see revised Fig. 5C).

Reviewer #3 (Comments to the Authors (Required)):

We would like to thank the reviewer for their supporting comments and constructive feedback. We have addressed all their minor comments as follows:

In the present manuscript Ozgencil et al. report the identification of a novel and replication stress dependent interaction between BRCA1 and the linker histone H1 and propose a role for histone H1 in replication fork stability. The interaction was identified in a previous mass spec from the same authors and proposed to be dependent on CK2 phosphorylation. However, mutation of BRCA1 on this specific residue does not abrogate this interaction, suggesting the presence of multiple interaction sites. Consistent with a functional role for this interaction silencing of histone H1 by siRNA prevented BRCA1 localization to sites of DNA damage while not preventing γ H2AX phosphorylation. Given the previously established link between BRCA1 and fork protection they show that loss of H1 induces fork degradation, all dependent on the MRE11 nuclease.

The work is interesting given the increased attention to histone H1 and its roles in regulating chromatin structure. The manuscript is well written and data are clearly presented. I have a few minor comments that authors should address to fully support publication.

It would be good to further strengthen the finding that this interaction and function does not depend on a specific histone H1 subtype. I understand that this issue is complicated by the different expression levels (in different cell lines) of the H1 variants but authors could strengthen this by:

1) analyse the interaction between BRCA1 and another histone H1 subtype, different from H1.3, at the endogenous levels (and/or at least in a different cell line ?).

We thank the reviewer for the experimental suggestion. We now provided clear evidence that BRCA1 interacts at the endogenous level also with H1.2 by both co-immunoprecipitation experiments and PLA assay (revised Fig. 1F and S1C).

2) The authors showed that expression of siRNA resistant GFP-H1.3 rescue BRCA1 recruitment. Could they see a similar effect with another histone H1 gene overexpression? this would strengthen their finding that this effect is not H1.3-specific.

To address this reviewer comment we generated U2OS cell lines stably expressing siRNA-resistant GFP-H1.1 and GFP-H1.4 and showed that expression of GFP-H1.1 and GFP- H1.4 (upon silencing of endogenous H1) is able to rescue BRCA1 chromatin recruitment. This data points to a non H1.3-specific mechanism for BRCA1 recruitment to stalled replication forks.

3) IP with flag 1.4 seems weaker. Is this a wb exposure issue? a plasmid expression problem or might depend on different properties of H1.4 ?

To address the reviewer comment we repeated co-immunoprecipitation experiments with Flag H1.4 and show a more clear WB in revised Fig. S1B. Of note, we now also show that overexpression of GFP-H1.4 rescues BRCA1 recruitment to stalled replication forks upon silencing of endogenous H1, pointing to a functional interaction between H1.4 and BRCA1.

4) are there other potential CK2 sites involved which might explain authors results? is this interaction reduced by CK2 inhibition for example? (this should be at least discussed)

We appreciate the reviewer's comment. To strengthen our manuscript and address the reviewer comment (as well as reviewer 2 comment) we have now further investigated the mechanistic basis of the H1-BRCA1 interaction. In particular, our new data suggest that interaction between BRCA1 and histone H1 requires both the C-terminal domain (containing the CK2 phosphorylation site S1336) and the ring domain of BRCA1 (revised Fig. S1A). These data suggest that interaction between H1 and BRCA1 involves multiple sites and/or indirect partners (e.g. BARD1). In addition to this, we also show that silencing of RNF8 strongly decrease the HU-dependent interaction between BRCA1 and H1, pointing to an important role for H1 ubiquitination in regulating BRCA1 recruitment to stalled replication forks. We discussed these results in the revised version of our manuscript.

5) Minor comment: Fig 2b. Contrast should be increased as gH2AX staining is hardly visible.

We thank the reviewer for the comment. We have now increased the contrast of γ H2AX staining in the revised version of our manuscript.

June 7, 2023

RE: Life Science Alliance Manuscript #LSA-2023-01933R

Dr. Roberto Bellelli
Queen Mary University of London
Barts Cancer Institute
Charterhouse Square
London EC1M 6BQ
United Kingdom

Dear Dr. Bellelli,

Thank you for submitting your revised manuscript entitled "The linker histone H1-BRCA1 axis is a crucial mediator of replication fork stability". We would be happy to publish your paper in Life Science Alliance pending final revisions necessary to meet our formatting guidelines.

- please add ORCID ID for the corresponding (and secondary corresponding) author--you should have received instructions on how to do so
- please upload all figure files as individual ones, including the supplementary figure files; all figure legends should only appear in the main manuscript file
- please add your main and supplementary figure legends to the main manuscript text after the references section
- please ensure the authors' names are consistent between the manuscript file and the system (e.g., Zuzana Hořejší in the manuscript file should match Zuzana Horejzi in the system).
- please add the Twitter handle of your host institute/organization as well as your own or/and one of the authors in our system
- please add callouts for Figures 1C and 5A to your main manuscript text
- please add a Summary Blurb/Alternate Abstract to our system
- please consult our manuscript preparation guidelines <https://www.life-science-alliance.org/manuscript-prep> and make sure your manuscript sections are in the correct order
- please add an Author Contributions section to your main manuscript text
- please add a conflict of interest statement to your main manuscript text

Figure checks:

- Scale bars are needed for Figures 1F, 2B, 2C, 3A, and S1D, and please indicate the size in the appropriate figure legend
- please add sizes next to all blots

A. FINAL FILES:

- An editable version of the final text (.DOC or .DOCX) is needed for copyediting (no PDFs).
- High-resolution figure, supplementary figure and video files uploaded as individual files: See our detailed guidelines for preparing your production-ready images, <https://www.life-science-alliance.org/authors>
- Summary blurb (enter in submission system): A short text summarizing in a single sentence the study (max. 200 characters)

including spaces). This text is used in conjunction with the titles of papers, hence should be informative and complementary to the title. It should describe the context and significance of the findings for a general readership; it should be written in the present tense and refer to the work in the third person. Author names should not be mentioned.

B. MANUSCRIPT ORGANIZATION AND FORMATTING:

Sincerely,

June 13, 2023

RE: Life Science Alliance Manuscript #LSA-2023-01933RR

Dr. Roberto Bellelli
Queen Mary University of London
Barts Cancer Institute
Charterhouse Square
London EC1M 6BQ
United Kingdom

Dear Dr. Bellelli,

Thank you for submitting your Research Article entitled "The linker histone H1-BRCA1 axis is a crucial mediator of replication fork stability". It is a pleasure to let you know that your manuscript is now accepted for publication in Life Science Alliance. Congratulations on this interesting work.

DISTRIBUTION OF MATERIALS:

Again, congratulations on a very nice paper. I hope you found the review process to be constructive and are pleased with how the manuscript was handled editorially. We look forward to future exciting submissions from your lab.

Sincerely,
